# Rethinking Individual Global Max in Cooperative Multi-Agent Reinforcement Learning

**Yitian Hong**
East China University of Science and Technology
ythong1314@mail.ecust.edu.cn

**Yaochu Jin**[*]
Bielefeld University
yaochu.jin@uni-bielefeld.de

**Yang Tang**[*]
East China University of Science and Technology
yangtang@ecust.edu.cn

## Abstract

In cooperative multi-agent reinforcement learning, centralized training and decentralized execution (CTDE) has achieved remarkable success. Individual Global Max (IGM) decomposition, which is an important element of CTDE, measures the consistency between local and joint policies. The majority of IGM-based research focuses on how to establish this consistent relationship, but little attention has been paid to examining IGM's potential flaws. In this work, we reveal that the IGM condition is a lossy decomposition, and the error of lossy decomposition will accumulated in hypernetwork-based methods. To address the above issue, we propose to adopt an imitation learning strategy to separate the lossy decomposition from Bellman iterations, thereby avoiding error accumulation. The proposed strategy is theoretically proved and empirically verified on the StarCraft Multi-Agent Challenge benchmark problem with zero sight view. The results also confirm that the proposed method outperforms state-of-the-art IGM-based approaches.

## 1 Introduction

Cooperative multi-agent reinforcement learning (MARL) has been proposed for multi-agent collaborations to accomplish many challenging tasks [1, 2, 3, 4]. MARL often relies on decentralized structures because of constraints in communication and observation commonly seen in applications [5]. Using additional information during the training process is a popular paradigm for decentralized MARL, known as centralized training [6, 7].

The value decomposition (VD) method [8], as one of the centralized training and decentralized execution (CTDE) paradigm, decomposes the joint-action value into multiple individual-action values and has achieved the state-of-the-art performance in StarCraft Multi-Agent Challenge (SMAC) [9]. In the CTDE paradigm, Individual Global Max (IGM) is an important principle in VD for efficiently facilitating centralized training for decentralized execution.

IGM was proposed in QMIX [10], a popular VD method, which uses a hypernetwork (MIX network) structure with additional environmental information to decompose the joint-action value into

---

[*]Corresponding author

This work was supported by the National Natural Science Foundation of China (Basic Science Center Program: 61988101), Natural Science Foundation of China (62136003, 62233005), the Programme of Introducing Talents of Discipline to Universities (the 111 Project) under Grant B17017 and Shanghai AI Lab.
Y. Jin is supported by an Alexander von Humboldt Professorship for AI endowed by the German Federal Ministry of Education and Research.

36th Conference on Neural Information Processing Systems (NeurIPS 2022).

individual-action values. However, the structure of the mixing network in QMIX assumes that the joint-action value and individual-action values are strictly monotonic, which is, however, only a sufficient condition for IGM. To achieve error-free VD, however, a necessary and sufficient condition for IGM is required. Following the idea of QMIX, a series of research has been reported to improve the performance of QMIX by constructing more sophisticated mixing network structures. For instance, Yang et al. [11] present a framework, called Qatten, which introduces an attention model into the mixing network to accelerate the training process. QPLEX, proposed by Wang et al. [12], is constructed using a duplex dueling network that satisfies the necessary and sufficient condition for IGM. DMIX [13] integrates value function factorization methods into distributed reinforcement learning for highly stochastic environments. Surprisingly, recent results show that QMIX with fine-tuned hyperparameters and normalization outperforms many other recently developed methods on the SMAC benchmark problem [14].

To the best of our knowledge, not much work has been dedicated to examining potential defects of IGM. In this paper, we prove that the IGM cannot equivalently transform actions from global state dependence to local observation dependence. In other words, the decomposition from global action value to individual action values is lossy. Furthermore, we point out that the error will accumulate in the reinforcement learning training process. As a result, the error of lossy decomposition can be amplified, which greatly limits the use of IGM decomposition. Hence, hypernetwork-based VD methods suffer from a significant performance degradation.

To address the aforementioned lossy decomposition problem, this paper proposes a novel training paradigm, called DAgger-based IGM (IGM-DA). Since lossy decomposition is unavoidable due to limited perception, this work aims to prevent the lossy decomposition error from being accumulated, considering that error accumulation mainly occurs in the reinforcement learning training process. To this end, we propose a novel training paradigm consisting an IGM decomposition training process and an imitation learning training process. The former trains a global observation MARL agent as an expert, whilst the latter uses an imitation learning technique, namely DAgger [15], to decompose the trained result in the former process relying on global observation into the latter that replies on local observation only.

In the next section, we theoretically prove that the lossy decomposition error accumulates in hypernetwork-based VD methods and that error accumulation can be avoided using the proposed IGM-DA. To empirically validate the effectiveness of IGM-DA in mitigating the influence of the lossy decomposition, we investigate the performance change when the perception range in SMAC varies. Ablation studies and additional analysis confirm the importance of the proposed IGM-DA by showing that the performance is significantly enhanced when IGM-DA is embedded in QMIX, QPLEX, and DMIX, three state-of-the-art IGM algorithms.

## 2 Analysis

As shown in previous work [16, 17], the cooperative MARL problem can be modeled as a Decentralized Partially Observable Markov Decision Processes (DEC-POMDPs). DEC-POMDPs are defined by a tuple of $(S, Z, O, T, U, P, r, N, \gamma)$, where $s \in S$ denotes the current state of the environment. At time instant $t$, each agent $i \in N \equiv \{1, ..., n\}$ takes actions $u_i \in U$ to facilitate cooperation. All these actions form a joint action set $\boldsymbol{u} \in \boldsymbol{U} \equiv U^n$. $P(s'|s, \boldsymbol{u}) : S \times \boldsymbol{U} \times S \to [0, 1]$ represents the state transition after the agents take the joint action. $r(s, \boldsymbol{u}) : S \times \boldsymbol{U} \to \mathbb{R}$ denotes the reward shared by all agents and $\gamma \in [0, 1)$ is the discount factor.

In a *partial observation* setting, each agent can only perceive local information of environment $z \in Z$ according to the observation function $O(s, i) : S \times N \to Z$. The action-observation history for each agent is $\tau^i \in T \equiv (Z \times U)$. According to action-observation history, each agent conditions its own strategy $\pi^i(u^i|\tau^i) : T \times U \to [0, 1]$. Based on the joint strategy $\pi$, the *joint-action value function* $Q^\pi(s_t, \boldsymbol{u_t}) = \mathbb{E}_{s_{t+1:\infty}, \boldsymbol{u}_{t+1:\infty}}[\sum_{k=0}^\infty \gamma^k r_{t+k}|s_t, \boldsymbol{u_t}]$ can be established. Furthermore, finding the optimal strategy can be reduced to finding the optimal joint *action-value* function $Q^*$.

### 2.1 Individual Global Max (IGM)

In the centralized training process, each agent can access additional global information for strategy training. On the other hand, during the decentralized execution process, each agent can only access

its own local action-observation history $\tau^i$ for decision making. This paradigm, known as CTDE, is widely used in cooperative MARL. IGM is an important principle for realizing CTDE in value-based MARL methods, which can be represented as follows:

$$\arg\max_{\boldsymbol{u}} Q_{tot}(s, \boldsymbol{u}) = \begin{pmatrix} \arg\max_{u^1} q_1\left(\tau^1, u^1\right) \\ \vdots \\ \arg\max_{u^n} q_n\left(\tau^n, u^n\right) \end{pmatrix}, \tag{1}$$

where the individual agent utility is represented by $q_i, i \in N$. The IGM principle affirms the consistency of global and local action selection, as well as the factorization relationship between the *joint-action-value function* $Q_{tot}$ and the *local-action-value function* $q_i$.

**Assumption 2.1** *IGM decomposition is realized by introducing the IGM principle into hypernetwork construction. As shown in Figure 1, IGM MIX represents a meticulously designed network enabling to approximately learn the joint-action-value $Q_{tot}$ relying on global observation from the action-values of individual agents $q_i$ based on local observations. In this case, the learned $Q_{tot}$ is denoted by $IGM(q_1, ..., q_n)$.*

The hypernetwork in IGM based decomposition is currently one of the most popular structures, for example, in AVD-Net [18], MAVEN [19], and ROMA [20]. All these methods integrate the IGM consistency condition, which is achieved through the delicate design of the IGM MIX network, into the network structure, thereby eliminating possible inconsistencies between the local maximum *action-value* and the global maximum *joint-action value*.

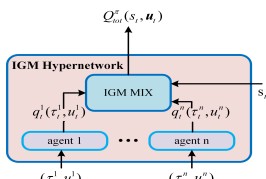

To update the *joint-action value*, the Bellman equation [21] is introduced. Then, the *joint-action value* function is reformulated as follows:

Figure 1: The hypernetwork of IGM decomposition, where each agent holds local-observation input.

$$\begin{aligned} Q_{tot}^\pi(s_t, \boldsymbol{u}_t) &= IGM(q_1(\tau_t^1, u_t^1), ..., q_n(\tau_t^n, u_t^n)) \\ &= r + \gamma\max_{\boldsymbol{u}_{t+1}} \left(IGM(q_1(\tau_{t+1}^1, u_{t+1}^1), ..., q_n(\tau_{t+1}^n, u_{t+1}^n))\right). \end{aligned} \tag{2}$$

Benefiting from the hypernetwork of the IGM decomposition, the global maximum *joint-action-value* function $IGM(q_1, ..., q_n)$ can be ensured by maximizing the local *action-value* function $[q_i(\tau_{t+1}^i, u_{t+1}^i)]_{i=1}^n$. Thus, a new iterative equation of the local *action-value* is obtained as follows:

$$IGM(q_1(\tau_t^1, u_t^1), ..., q_n(\tau_t^n, u_t^n)) = r + \gamma IGM(\max_{u_{t+1}^1} q_1(\tau_{t+1}^1, u_{t+1}^1), ..., \max_{u_{t+1}^n} q_n(\tau_{t+1}^n, u_{t+1}^n)). \tag{3}$$

This way, each agent participates in the estimation of *joint-action-value* to improve the efficiency of exploration. Therefore, the hypernetwork in IGM makes it possible to extend the *action-value* iteration equation from a global state and joint action selection to local scenarios and individual action selections.

## 2.2 Defects of IGM

### 2.2.1 The Lossy Decomposition

The IGM factorization consists of the following two steps:

$$\begin{cases} \arg\max_{\boldsymbol{u}} Q_{tot}(s, \boldsymbol{u}) = \begin{pmatrix} \arg\max_{u^1} Q_1\left(s, u^1\right) \\ \vdots \\ \arg\max_{u^n} Q_n\left(s, u^n\right) \end{pmatrix} \\ \begin{pmatrix} \arg\max_{u^1} Q_1\left(s, u^1\right) \\ \vdots \\ \arg\max_{u^n} Q_n\left(s, u^n\right) \end{pmatrix} = \begin{pmatrix} \arg\max_{u^1} q_1\left(\tau^1, u^1\right) \\ \vdots \\ \arg\max_{u^n} q_n\left(\tau^n, u^n\right) \end{pmatrix} \end{cases}. \tag{4}$$

$Q_i(s, u^i)$ represents the local *action-value* based on the global state rather than the local observation. The first line in equation 4 indicates that for the same state, *joint-action-value* $Q_{tot}(s, \boldsymbol{u})$ is decomposed into multiple *local-action-values* $Q_i(s, u^i)$. Thus, individual action selection does not need to rely on the policy of others. This also makes each individual capable of independent exploration, which is the fundamental goal of the VD method. The second line in equation 4 indicates that for local action selection, the global state-based *action-value* is converted into local observation-based. Therefore, the decentralized execution process can rely on individual local observations only. Compared with equation 1, equation 4 indicates two roles of the IGM decomposition: decoupling interdependence between actions of different agents and converting actions from global state dependency into local observation dependency. In this work, we focus on the second part, i.e., how to more accurately learn the local action-values based on the action-value of individual agents with global observation.

Existing work on hypernetwork based VD, such as QMIX [10], focuses on increasing the hypernetwork learning potential by introducing global states into the hypernetwork, without paying much attention to the dependence of actions on global state. In the following, we will show that *action-value* based on the global state cannot be perfectly decomposed into action-values based on local observation only. In other words, the decomposition based on IGM is lossy when the local observation is insufficient. The definition of insufficient observation and lossy decomposition is given as follows:

**Definition 1** *(Insufficient Observation). Local observation $\tau$ is an insufficient observation of global state $s$, if there is a case where global state $s$ changes while local observation $\tau$ does not change.*

**Definition 2** *(Lossy Decomposition). For any individual action-value functions based local observation $[q_i(\tau^i, u^i) : T \times U \to \mathbb{R}]_{i=1}^n$. The decomposition from $Q_{tot}(s, \boldsymbol{u})$ into $[q^i(\tau^i, u^i)]_{i=1}^n$ is lossy, if $\exists \, s \in S, \tau^i \in T$, s.t. $\arg\max\limits_{\boldsymbol{u}} Q_{tot}(s, \boldsymbol{u}) \neq [\arg\max\limits_{u^i} q_i(\tau^i, u^i)]_{i=1}^n$*

This paper also gives the proposition of the existence of lossy decomposition and a proof is given in Appendix A.

**Proposition 1** *(Existence of Lossy Decomposition). Let $\tau$ be an insufficient observation of global state $s$ as defined in 1, then $\exists \, Q_{tot}(s, \boldsymbol{u})$ such that the decomposition from $Q_{tot}(s, \boldsymbol{u})$ into $[q^i(\tau^i, u^i)]_{i=1}^n$ is lossy.*

As a consequence, for the same observation $\tau^i$, the agents cannot distinguish whether the environmental state outside their sensing range has changed or not. Because the joint action selection of the agents is based on global information $s$, individual agents select the action based only on their local observations $\tau^i$, resulting in incorrect action selection (*lossy decomposition*).

To mitigate the negative impact of partial observations, global information can be introduced in the hypernetwork during training progress (CTDE). As shown in Figure 1, the global information $s$ is considered as an embedded input in the hypernetwork-based method to prevent its direct impact on action selection. Therefore, the action selection module based on local observation can be separated more conveniently. Namely, the global information is kept from influencing the individual action selection, resulting in a *lossy decomposition*. In addition to the proof in Appendix A, the influence of *lossy decomposition* on the performance will be further shown in discussing the experimental results.

### 2.2.2 Error Accumulations

The sensing limit requires that the learned strategies are based only on local information. Therefore, in most cases, lossy decomposition in IGM is inevitable. In addition to lossy decomposition, we find that IGM also suffers from the problem of error accumulation during the training process.

Because IGM is a lossy decomposition, equation 2 can be rewritten by:

$$Q_{tot}^\pi(s_t, \boldsymbol{u}_t) \approx IGM(q_1(\tau_t^1, u_t^1), ..., q_n(\tau_t^n, u_t^n)), \tag{5}$$

where the approximately equal symbol signifies the existence of lossy decomposition. Similarly, equation 3 can be revised as follows:

$$IGM(q_1(\tau_t^1, u_t^1), ..., q_n(\tau_t^n, u_t^n)) \approx r + \gamma IGM(\max\limits_{u_{t+1}^1} q_1(\tau_{t+1}^1, u_{t+1}^1), ..., \max\limits_{u_{t+1}^n} q_n(\tau_{t+1}^n, u_{t+1}^n)).$$
$$\tag{6}$$

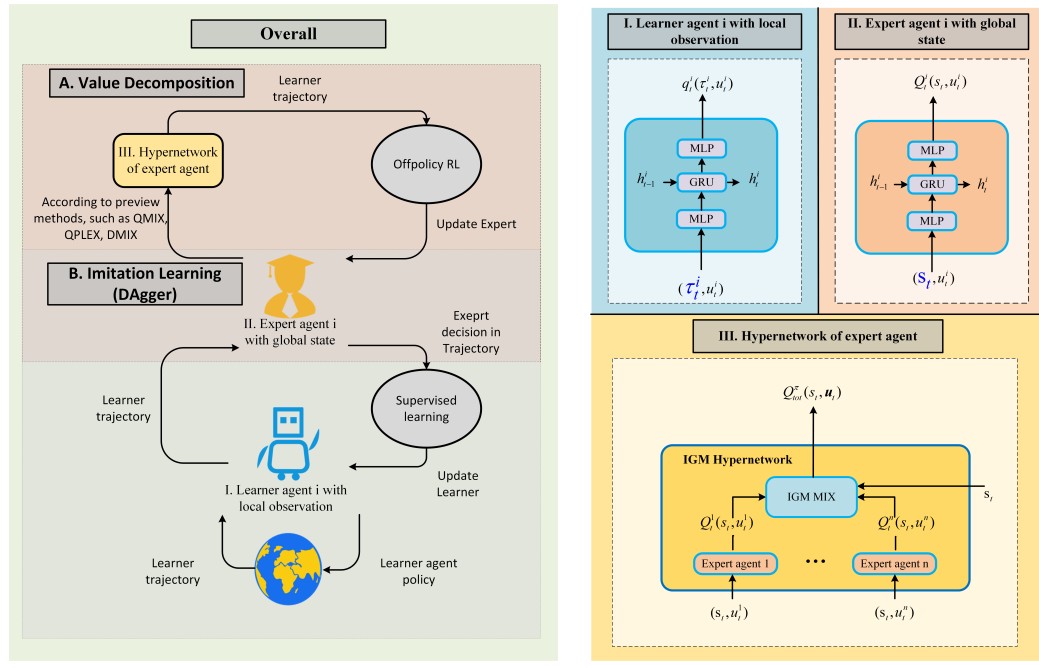

(a) The proposed IGM-DA framework.  (b) The detailed network structure.

Figure 2: The proposed IGM-DA framework and detailed network structure. (a) The value decomposition part (the upper part) trains an individual expert agent with global state; the imitation learning part (the lower part) trains an individual learner agent with local observation through supervised learning. (b) The detailed network structure of learner agent, expert agent, and hypernetwork of expert agent.

In the following, we will show that errors resulting from the lossy decomposition will accumulate in the iterative training. Let $error_{dec}$ denote the error generated by lossy decomposition in IGM, $error_{other}$ the remaining errors caused e.g., by noisy input information and limited network learning ability, and $Error(Q) = Q - \hat{Q}$ is the total error in the training process, where $Q$ is the true action value, and $\hat{Q}$ is the calculated action value. Then we have the following Proposition.

**Proposition 2** *(IGM with error accumulation).*
*According to assumption 2.1, the IGM decomposition is implemented by incorporating the IGM principle into the hypernetwork construction. Then $Q_{tot}$ is updated by the Bellman equation, thus updating $[q_i(\tau_t^i, u_t^i)]_{i=1}^n$ according to equation 5. The total error can be expressed by:*

$$Error(Q_{tot}^\pi(s_t, \boldsymbol{u}_t)) = \sum_{i=t}^{done} \gamma^{i-t}[error_{dec}(i) + error_{other}(i)]. \tag{7}$$

Error accumulation is proved in Appendix A. In the following, we introduce imitation learning based on the DAgger structure into IGM, called IGM-DA, to avoid error accumulation.

## 3 Method

### 3.1 Error-accumulation-free IGM

Due to the error accumulation in the training process, the total error of the whole system may become larger and larger through the training process. To resolve this problem, we propose an IGM-DA training paradigm to prevent error accumulation.

In the first stage, as shown in Figure 2(a), we obtain individual expert agents based on the global state through RL training of the IGM hypernetwork (the upper part). In the second stage, we train the

individual learner agent based on local observation by means of imitation learning (the lower part). This way, lossy decomposition is separated from the iterative training process so that it occurs in the second stage only.

Figure 2(b) shows the detailed network structure of the learner agent, expert agent, and hypernetwork of expert agent. In the hypernetwork, the IGM MIX acts as a connection between action-values of individual expert agents $[q_i(s_t^i, u_t^i)]_{i=1}^n$ with global state (instead of $[q_i(\tau_t^i, u_t^i)]_{i=1}^n$ with local observation) and joint-action-values $Q_{tot}$. Thus, equation 5 is rewritten by:

$$Q_{tot}^\pi(s_t, \boldsymbol{u}_t) = IGM(q_1(s_t^1, u_t^1), ..., q_n(s_t^n, u_t^n)), \tag{8}$$

Equation 8 also corresponds to the first line in equation 4. Because both the left and right sides of the equations are dependent on global state, lossy decomposition caused by insufficient observation does not exist.

**Proposition 3** *(IGM Integrated imitation learning without error accumulation).*
*If $Q_{tot}$ is updated by the Bellman equation, thus updating $[q_i(s_t^i, u_t^i)]_{i=1}^n$ according to equation 8. then $[q_i(\tau_t^i, u_t^i)]_{i=1}^n$ is obtained from $[q_i(s_t^i, u_t^i)]_{i=1}^n$ by supervised imitation learning. Thus, equation 7 can be rewritten as follows:*

$$Error(Q_{tot}^\pi(s_t, \boldsymbol{u}_t)) = \sum_{i=t}^{done} \gamma^{i-t}[error_{other}(i)] + error_{dec}(t). \tag{9}$$

By comparing equation 7 and equation 9, we can see that error accumulation resulting from lossy decomposition can be avoided. The proof of Proposition 3 is provided in Appendix A.

## 3.2 Imitation Learning and Integrated DAgger

As shown in the Figure 2, we use imitation learning to avoid error accumulation. Common imitation learning, represented by Behavioral Cloning [22] and DAgger [23], aims to learn strategies from expert experience. Different from traditional imitation learning, experts in the proposed IGM-DA are not real experts but an intermediate learning result of RL. The main difference between expert agents and learner agents lies in the fact that expert agents can observe the whole state of the environment, while learner agents can only observe within a limited range. Note, however, that Behavioral Cloning is not well suited for the present work since in Behavioral Cloning the virtual experts modify their strategies through reinforcement learning based on the data collected by themselves, which remain to be global information based. By contrast, the virtual experts in DAgger can constantly modify their strategies through reinforcement learning based on the data collected by the learners. This way, the policies learned by the virtual experts based on global information can be adapted to the local observations.

Recall that expert strategies can observe global information, while learners can only observe local information. Although we have already theoretically analyzed that the integrated imitation learning structure in Figure 2 can prevent the error accumulation, we still need to find a proper decomposition in the imitation learning stage to reduce the error. Similar to Proposition 1, proposition 4 can be obtained:

**Proposition 4** *(Existence of Lossy Decomposition 2). Let $\tau$ be an insufficient observation of global state $s$ as defined in 1. Then $\exists [q^i(s^i, u^i)]_{i=1}^n$ such that the decomposition from $[q^i(s^i, u^i)]_{i=1}^n$ to $[q^i(\tau^i, u^i)]_{i=1}^n$ is lossy.*

It is worth noting that $q^i(\tau^i, u^i)$ represents the strategy based on local observation, and $P_\pi(u^i|s)$ represents the expert strategy based on global observation learned:

$$P_\pi(u^i|s) = \begin{cases} 1 & \text{if } u^i = \underset{u^i}{\arg\max}\, q^i(s, u^i) \\ 0 & \text{if } u^i \neq \underset{u^i}{\arg\max}\, q^i(s, u^i) \end{cases}. \tag{10}$$

In order to find the optimal decomposition, we need to define the optimal decomposition from $[q^i(s^i, u^i)]_{i=1}^n$ to $[q^i(\tau^i, u^i)]_{i=1}^n$. To this end, we introduce the Bayesian expected loss [24] in lossy imitation learning.

**Proposition 5** (*action-value after lossy imitation learning*). *Suppose we have $k$ samples that satisfy local observation $\tau$. Then the optimal action-value after imitation learning will be:*

$$q^i(\tau^i, u^i) = 1/k \sum_s [P_\pi(u^i|s)], \tag{11}$$

Proposition 5 is proved in Appendix A. The loss function of the supervised imitation learning is $loss = q^i(\tau^i, u^i) - 1/k \sum_s [P_\pi(u^i|s)]$.

## 4 Experimental Results

To rigorously investigate the performance of the proposed learning strategy, we adopt the StarCraft II benchmark task as the test problem. Partial observations of the environment are reflected in the limited sensing ranges of the agents, resulting in lossy decomposition. In this section, we first show the existence of lossy decomposition before comparing the proposed framework with state-of-the-art baselines, including QMIX, QPLEX, and DMIX. To clearly show the robustness of the proposed framework, we set the vision range of the agents in the environment to 0. In this case, each agent only knows its own attributes and optional actions. Several ablation studies are also given by comparing the proposed framework with DAgger. The implementation details and experimental settings can be found in Appendices C and D. For fair evaluations, the hyper-parameters of all algorithms under comparison as well as the optimizers, are the same, and the experimental results are presented with the average performance with 25-75% percentile. Moreover, the presented curves are smoothed by a moving average filter with its window size being set to 5 for better visualization. The code of the proposed algorithm can be downloaded at [*].

### 4.1 Lossy decomposition

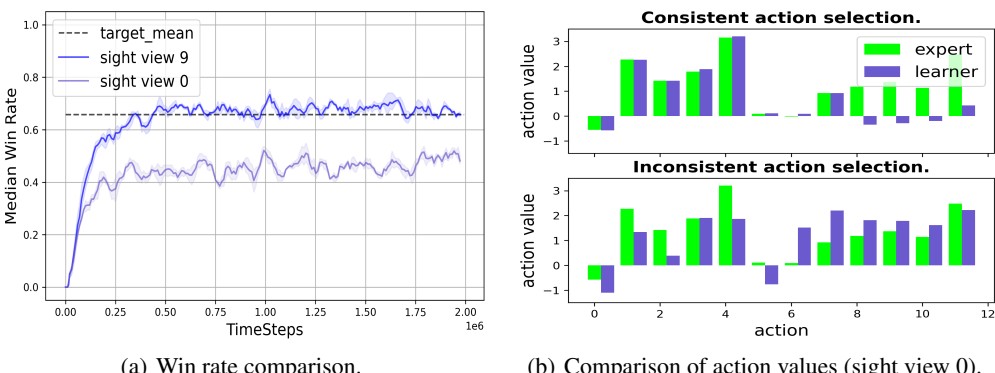

(a) Win rate comparison.          (b) Comparison of action values (sight view 0).

Figure 3: Verification of the existence of lossy compression. Because multiple global states may correspond to the same local observation, the action value under a local observation is the weighted average of the action values of multiple global states.

In order to verify the existence of lossy decomposition in StarCraft II, we first train a group of agents in a relatively large sensing range, which is called sight-view hereafter (sight-view 9), and then use the agent with sight-view 0 to imitate the trained policies. As shown in Figure 3 (a), *target_mean* represents the average winning rate of a set of strategies trained under sight-view 9. Other curves plot the imitation results of the trained strategy (expert action value) under different settings. *sight view 9* denotes the sensing (viewing) range of the learner agents is 9. Similarly, *sight view 0* denotes the viewing range is 0. Figure 3 (b) shows the distribution comparison of action values for sight view 0. Because of the insufficient observation, the distribution of action value will have errors that lead to the final wrong action selection. Compared to the experimental results of different fields of vision, it can be concluded that the strategy learned in sight view 9 depends on additional information beyond sight view 0.

---

[*]`https://github.com/momo-xiaoyi/pymarl_HDA`

## 4.2 Robustness to zero sight view

The environments in SMAC are divided into three difficulty levels: Easy, Hard, and Super Hard. In this work, the scope of the agents' vision is limited to 0, which poses additional difficulty to the environment. We choose six of these environments for performance evaluation: (a) 3s5z(Easy), (b) 5m_vs_6m(Hard), (c) MMM2(Super Hard), (d) 8m(Easy), (e) 3s_vs_5z(Hard), and (f) 8m_vs_9m(Hard). It is worth noting that because of the increased difficulty, the win rate of all test algorithms may be 0 in a Super Hard environment. In order to better show the differences between algorithms, we do not pay much attention to the Super Hard environments.

In Figure 4, we use *-DA* to represent variants of the original method combined with the *IGM-DA* framework. We use coarse curves for the *IGM-DA* results to make it easier to distinguish. We find that *IGM-DA* outperforms the other compared algorithms. The detailed results of Figure 4 are listed in Table 1. The best performing algorithms in different environments are as follows: 3s5z (qmix-DA); 5m_vs_6m (dmix-DA); MMM2 (qmix-DA,qplex-DA); 8m (dmix-DA); 3s_vs_5z (qmix-DA); 8m_vs_9m (qmix-DA). The average success rate of the proposed approach is improved by about 20% over that of the compared algorithms. Figure 5(c) shows the robustness of our method for different sight views.

Table 1: Average win rate of SMAC challenges in sight view 0

|  | 3s5z | 5m_vs_6m | MMM2 | 8m | 3s_vs_5z | 8m_vs_9m | Avg.Score |
|---|---|---|---|---|---|---|---|
| qmix | 90.2 | 36.4 | 26.2 | 98.3 | 1.9 | 33.9 | 47.8 |
| qmix-DA | **93.3** | **50.6** | **55.4** | **99.2** | **43.9** | **70.8** | **68.9** |
| qplex | 87.0 | 5.0 | 0.0 | 98.7 | 18.9 | 36.2 | 41.0 |
| qplex-DA | **88.3** | **38.9** | **55.4** | **99.6** | **31.6** | **64.8** | **63.1** |
| dmix | 76.2 | 58.7 | 0.0 | 99.4 | 3.3 | **64.7** | 50.4 |
| dmix-DA | **91.7** | **59.0** | **46.2** | **99.8** | **40.6** | 64.4 | **67.0** |

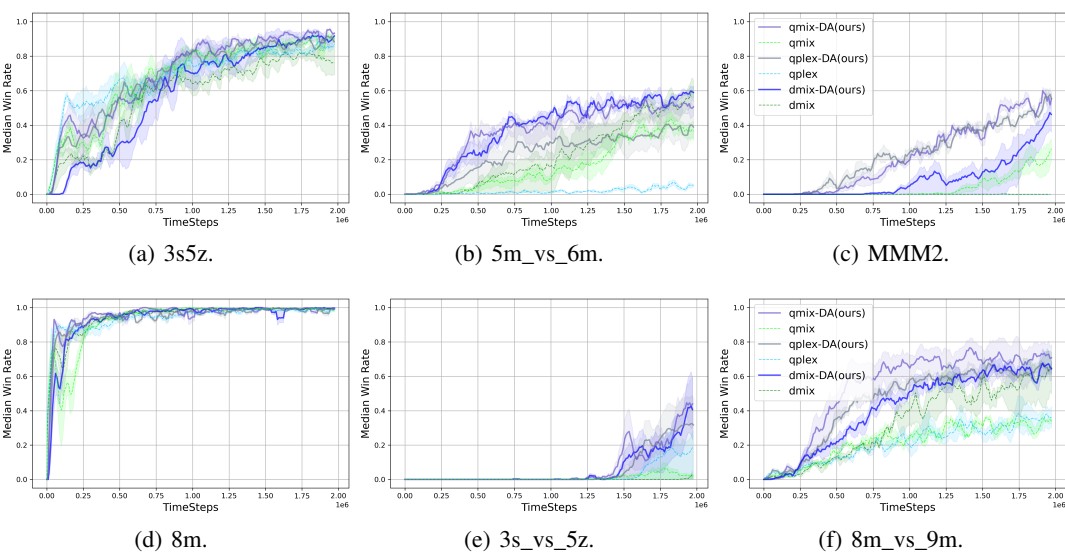

Figure 4: Results of QMIX, QPLEX and DMIX with or without integrated DAgger in six environments, showing that integrated DAgger can significantly increase the median win rate in 0 sight view.

## 4.3 Ablation Studies

Figure 5 (a) and (b) plots the comparative results of different imitation learning structures, where *DA* refers to DAgger, and *BC* refers to Behavior Cloning. A detailed introduction of imitation learning is given in Appendix B. As shown in Figure 5(a), when combined with imitation learning,

the algorithm's performance can be significantly improved. The results in Figure 5(b) confirm the limitation of Behavior Cloning. We can clearly see that there is a huge gap between Behavioral Cloning and DAgger algorithm in the Super Hard MMM2 environments.

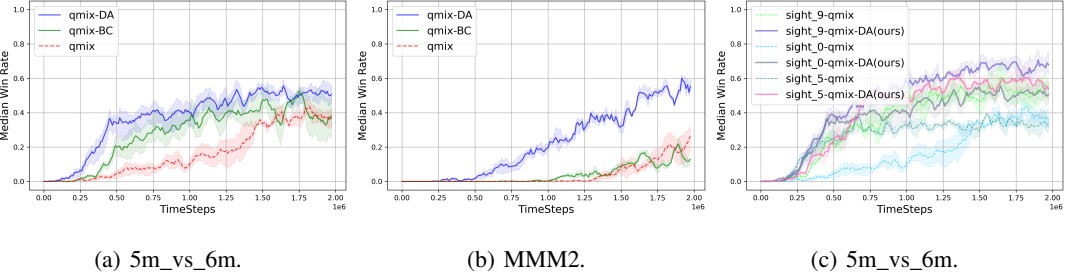

(a) 5m_vs_6m.       (b) MMM2.       (c) 5m_vs_6m.

Figure 5: (a) and (b) :Results comparing Behavioral Cloning and DAgger. (c) Our algorithm is robust in different sight views.

# 5    Related Work

The proposed IGM-DA aims to enhance the robustness of the value decomposition method in the presence of partial observation when no communications between the agents are allowed. In case communications are possible, the information loss might be properly compensated. For example, Foerster et al. [25] proposed deep reinforcement learning for communication topology learning, where agents can use information from others to stabilize training. Along the same line, two information-theoretic regularizers between value function factorization learning and communication learning were proposed in [26], which reduces the amount of required communication without sacrificing the performance. Chen et al. [27] considers the problem of inefficient sampling caused by frequent changes in communication channels, and proposes to accelerate communication learning by integrating centralized learning and knowledge distillation. Although Chen et al. also use fully centralized training, which is similar to this work, the motivations and assumptions are completely different. The reader may refer to [28] for more research on communication-based learning. Moreover, studies on the algorithmic property of the VD methods from different perspectives have also been reported. Through a large number of experiments in one-shot (i.e., non-sequential) problems, Castellini et al. [29] visualize the expression ability of various MARL methods. Factorized Multi-Agent Fitted Q-Iteration was proposed by Wang et al. [30] to analyze the cooperative MARL based on value decomposition. Under the IGM conditions, Huang et al. [31] take into account the sub-team coordination problem to achieve more efficient collaborations.

# 6    Conclusion and Future Work

In this paper, we have pointed out that IGM decomposition is a lossy decomposition, and that the error resulting from the lossy decomposition may accumulate in the training process. The accumulated error may seriously degrade the performance of VD-based algorithms. To tackle the above problems, we have proposed IGM-DA, which integrates imitation learning into IGM decomposition. We show theoretically and empirically that the proposed framework can prevent error accumulation by introducing imitation learning into the training process, making it possible to adapt the learned policies to the local information, thereby avoiding error accumulation.

The proposed work is able to achieve the best performance improvement in extreme environments of zero vision. If the sight range becomes larger, the improvement will be less significant (refer to the experimental results shown in Appendix F). In addition, this paper focuses on one class of value decomposition methods, namely hypernetwork with local observation. Other methods, such as QTRAN [32] based on loss design are beyond the scope of this paper. Finally, we have adopted a classical imitation learning technique for avoiding error accumulation. Therefore, an immediate future work is to integrate STOA imitation learning technologies with multiple value decomposition methods. Furthermore, we plan to extend the imitation learning structure to partially observed single-agent reinforcement learning algorithms.

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
