(a) 3s5z.  (b) 5m_vs_6m.  (c) MMM2.

(d) 8m.  (e) 3s_vs_5z.  (f) 8m_vs_9m.

Figure 4: Results of QMIX, QPLEX and DMIX with or without integrated DAgger in six environments, showing that integrated DAgger can significantly increase the median win rate in 0 sight view.

## 4.3 Ablation Studies

Figure 5 (a) and (b) plots the comparative results of different imitation learning structures, where *DA* refers to DAgger, and *BC* refers to Behavior Cloning. A detailed introduction of imitation learning is given in Appendix B. As shown in Figure 5(a), when combined with imitation learning,

the algorithm's performance can be significantly improved. The results in Figure 5(b) confirm the limitation of Behavior Cloning. We can clearly see that there is a huge gap between Behavioral Cloning and DAgger algorithm in the Super Hard MMM2 environments.

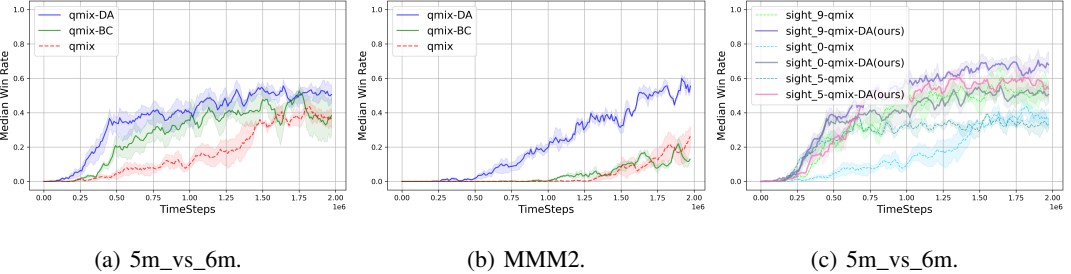

(a) 5m_vs_6m.       (b) MMM2.       (c) 5m_vs_6m.

Figure 5: (a) and (b) :Results comparing Behavioral Cloning and DAgger. (c) Our algorithm is robust in different sight views.

## 5 Related Work

The proposed IGM-DA aims to enhance the robustness of the value decomposition method in the presence of partial observation when no communications between the agents are allowed. In case communications are possible, the information loss might be properly compensated. For example, Foerster et al. [25] proposed deep reinforcement learning for communication topology learning, where agents can use information from others to stabilize training. Along the same line, two information-theoretic regularizers between value function factorization learning and communication learning were proposed in [26], which reduces the amount of required communication without sacrificing the performance. Chen et al. [27] considers the problem of inefficient sampling caused by frequent changes in communication channels, and proposes to accelerate communication learning by integrating centralized learning and knowledge distillation. Although Chen et al. also use fully centralized training, which is similar to this work, the motivations and assumptions are completely different. The reader may refer to [28] for more research on communication-based learning. Moreover, studies on the algorithmic property of the VD methods from different perspectives have also been reported. Through a large number of experiments in one-shot (i.e., non-sequential) problems, Castellini et al. [29] visualize the expression ability of various MARL methods. Factorized Multi-Agent Fitted Q-Iteration was proposed by Wang et al. [30] to analyze the cooperative MARL based on value decomposition. Under the IGM conditions, Huang et al. [31] take into account the sub-team coordination problem to achieve more efficient collaborations.

## 6 Conclusion and Future Work

In this paper, we have pointed out that IGM decomposition is a lossy decomposition, and that the error resulting from the lossy decomposition may accumulate in the training process. The accumulated error may seriously degrade the performance of VD-based algorithms. To tackle the above problems, we have proposed IGM-DA, which integrates imitation learning into IGM decomposition. We show theoretically and empirically that the proposed framework can prevent error accumulation by introducing imitation learning into the training process, making it possible to adapt the learned policies to the local information, thereby avoiding error accumulation.

The proposed work is able to achieve the best performance improvement in extreme environments of zero vision. If the sight range becomes larger, the improvement will be less significant (refer to the experimental results shown in Appendix F). In addition, this paper focuses on one class of value decomposition methods, namely hypernetwork with local observation. Other methods, such as QTRAN [32] based on loss design are beyond the scope of this paper. Finally, we have adopted a classical imitation learning technique for avoiding error accumulation. Therefore, an immediate future work is to integrate STOA imitation learning technologies with multiple value decomposition methods. Furthermore, we plan to extend the imitation learning structure to partially observed single-agent reinforcement learning algorithms.

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

# A Proof

**Proposition 1** *(Existence of Lossy Decomposition). Let local observation $\tau$ be a insufficient observation of global state $s$ as defined in 1. Then $\exists\, Q_{tot}(s, \boldsymbol{u})$, such that, the decomposition from $Q_{tot}(s, \boldsymbol{u})$ to $[q^i(\tau^i, u^i)]_{i=1}^n$ is lossy.*

*proof.* Since local observation $\tau$ is a insufficient observation of global state $s$, according to definition 1, there is a case where global state $s$ changes, but local observation $\tau$ does not change.

Suppose $s_1$ and $s_2$ are two global state before and after the change, the unchanged local observation is $\tau_*$. Corresponding to $s_1$ and $s_2$, the joint action value in different state is $Q_{tot}(s_1, \boldsymbol{u})$ and $Q_{tot}(s_2, \boldsymbol{u})$.

To prove the existence of lossy decomposition, we first assume the decomposition is lossless. Then, based on formula 1, we get :

$$\arg\max_{\boldsymbol{u}} Q_{tot}(s_1, \boldsymbol{u}) = [\arg\max_{u^i} q_i(\tau_*^i, u^i)]_{i=1}^n, \tag{12}$$

$$\arg\max_{\boldsymbol{u}} Q_{tot}(s_2, \boldsymbol{u}) = [\arg\max_{u^i} q_i(\tau_*^i, u^i)]_{i=1}^n. \tag{13}$$

We note that the latter part of formula 12 and 13 are the same, so we can reconstruct the formula as follows:

$$\arg\max_{\boldsymbol{u}} Q_{tot}(s_1, \boldsymbol{u}) = \arg\max_{\boldsymbol{u}} Q_{tot}(s_2, \boldsymbol{u}). \tag{14}$$

Formula 14 strictly requires the same optimal joint strategy under different states, which is impossible in practical applications. Therefore, the existence of lossy decomposition is proved through the counterevidence method.

**Proposition 2** *(IGM with error accumulation).*
*If $Q_{tot}$ is represented by $IGM(q_1(\tau_t^1, u_t^1), ..., q_n(\tau_t^n, u_t^n))$, the error will be accumulated:*

$$Error(Q_{tot}^{\pi}(s_t, \boldsymbol{u}_t)) = \sum_{i=t}^{done} \gamma^{i-t}[error_{dec}(i) + error_{other}(i)]. \tag{15}$$

We represent the error generated by lossy decomposition in IGM as $error_{dec}$, the remaining errors in IGM as $error_{other}$, the total error in the training process as $Error$.

*proof.* According to the training iteration equation 6, the current action value needs to use the action value at the next moment. Correspondingly, the training error at the next moment will be accumulated to the current moment. Similarly, the training error at the next next moment will be accumulated to the next moment. Thus, the current time error will continue to grow:

$$\begin{aligned} Error(Q_{tot}^{\pi}(s_t, \boldsymbol{u}_t)) &= error_{dec}(t) + error_{other}(t) + \gamma Error(Q_{tot}^{\pi}(s_{t+1}, \boldsymbol{u}_{t+1})) \\ &= error_{dec}(t) + error_{other}(t) + \gamma error_{dec}(t+1) + \gamma error_{other}(t+1) \\ &\quad + \gamma Error(Q_{tot}^{\pi}(s_{t+2}, \boldsymbol{u}_{t+2})) \\ &= \sum_{i=t}^{done} \gamma^{i-t}[error_{dec}(i) + error_{other}(i)]. \end{aligned}$$
$$\tag{16}$$

**Proposition 3** *(IGM Integrated imitation learning without error accumulation).*
*If $Q_{tot}$ is represented by $IGM(q_1(s_t^1, u_t^1), ..., q_n(s_t^n, u_t^n))$ and $[q_i(\tau_t^i, u_t^i)]_{i=1}^n$ is obtained from $[q_i(s_t^i, u_t^i)]_{i=1}^n$ by supervised imitative learning, the error accumulation of lossy decomposition will be prevented:*

$$Error(Q_{tot}^{\pi}(s_t, \boldsymbol{u}_t)) = \sum_{i=t}^{done} \gamma^{i-t}[error_{other}(i)] + error_{dec}(t). \tag{17}$$

*proof.* As shown in the upper part of Figure 2(a) and Figure 2(b), the hypernetwork of IGM is constructed to re-represent $Q_{tot}$. Unlike Figure 1, the current structure only decomposes the *joint action-value* $Q_{tot}(s, \boldsymbol{u})$ into multiple $Q_i(s, u^i)$ rather than $q_i(\tau^i, u^i)$. Since both are based on global information, there is no lossy decomposition caused by insufficient observation. In this case, the equivalent sign of equation 5 will be changed back to the equal sign as following:

$$Q_{tot}^\pi(s_t, \boldsymbol{u}_t) = IGM(Q_1(s_t, u_t^1), ..., Q_n(s_t, u_t^n)). \tag{18}$$

Similarly, equation 6 can be rewritten as follows:

$$IGM(Q_1(s_t, u_t^1), ..., Q_n(s_t, u_t^n)) = r + \gamma IGM(\max_{u_{t+1}^1} Q_1(s_{t+1}, u_{t+1}^1), ..., \max_{u_{t+1}^n} Q_n(s_{t+1}, u_{t+1}^n)). \tag{19}$$

As shown in Figure 2(a), the lower part uses imitation learning to realize the transition from $Q_i(s, u^i)$ to $q_i(\tau^i, u^i)$:

$$Q_i(s_t, u_t^i) \approx q_i(s_{t+1}, u_{t+1}^i). \tag{20}$$

As a result, the error generated by lossy decomposition in IGM , $error_{dec}$, only exists in the imitation learning process. We represent the error in the upper part as $Error_{upper}$, the error in the lower part as $Error_{lower}$. Then the total error in the training process is:

$$
\begin{aligned}
Error(Q_{tot}^\pi(s_t, \boldsymbol{u}_t)) &= Error_{upper}(Q_{tot}^\pi(s_t, \boldsymbol{u}_t)) + Error_{lower}(Q_{tot}^\pi(s_t, \boldsymbol{u}_t)) \\
&= error_{other}(t) + \gamma Error_{upper}(Q_{tot}^\pi(s_{t+1}, \boldsymbol{u}_{t+1})) + Error_{lower}(Q_{tot}^\pi(s_t, \boldsymbol{u}_t)) \\
&= error_{other}(t) + \gamma error_{other}(t+1) + \gamma Error_{upper}(Q_{tot}^\pi(s_{t+2}, \boldsymbol{u}_{t+2})) \\
&\quad + Error_{lower}(Q_{tot}^\pi(s_t, \boldsymbol{u}_t)) \\
&= \sum_{i=t}^{done} \gamma^{i-t}[error_{other}(i)] + error_{dec}(i).
\end{aligned}
\tag{21}
$$

Thus, the error of lossy decomposition is not accumulated.

**Proposition 5** *(action-value after lossy imitation learning). Suppose we have $k$ samples that satisfy local observation $\tau$. Then the optimal action-value after lossy imitative learning is:*

$q^i(\tau^i, u^i) = 1/k \sum_s [P_\pi(u^i|s)].$

*Proof.*

**Definition 3** *(expected loss in lossy imitation learning). For local observation $\tau$, the confidence of different global environments is $P(s|\tau)$, the strategy based on global state is $P_\pi(u^j|s)$. $\lambda_{ij}$ denotes the penalty function for misjudged from $u_j$ to $u_i$. The expected loss of action $u^i$ when local observation is $\tau$:*

$R(u^i|\tau) = \sum_j \sum_s [\lambda_{ij} P_\pi(u^j|s) P(s|\tau)].$

Because the action with the minimum expected loss is the most Valuable, the optimal modified *action-value* of local observation is $q^i(\tau^i, u^i) = -R(u^i|\tau)$. Let $\lambda_{ij}$ be a common penalty function as follows :

$$\lambda_{ij} = \begin{cases} 1 & if\ i \neq j \\ 0 & if\ i = j \end{cases}. \tag{22}$$

In other words, when the selected action is inconsistent with the expected action, it will produce a punishment.

With definition 3 and penalty function $\lambda_{ij}$ in equation 22, we can get the following:

$$
\begin{aligned}
R(u^i|\tau) &= \sum_j \sum_s [\lambda_{ij} P_\pi(u^j|s) P(s|\tau)] = \sum_{j \in \{j \neq i\}} \sum_s [P_\pi(u^j|s) P(s|\tau)] \\
&= \sum_s [P(s|\tau) \sum_{j \in \{j \neq i\}} [P_\pi(u^j|s)]] = \sum_s [P(s|\tau)[1 - P_\pi(u^i|s)]]
\end{aligned}
\tag{23}
$$

Because $P(s|\tau)$ is the confidence of different global environments when the current local observation is $\tau$, the probability of global state $P(s|\tau)$ can be can be obtained by sampling a large number of data. Based on the sampling theorem, the above formula is reconstructed as follows:

$$R(u^i|\tau) = \frac{1}{k} \sum_{s \in S_\tau} [1 - P_\pi(u^i|s)], \tag{24}$$

where $S_\tau$ denotes $k$ sampled data whose local observation is $\tau$. 1 is a constant we can leave out, then it ends up with:

$$R(u^i|\tau) = -\frac{1}{k} \sum_{s \in S_\tau} [P_\pi(u^i|s)], \tag{25}$$

Because the action with the minimum expected risk is the most popular, the modified action-value of local observation is

$$q^i(\tau^i, u^i) = -R(u^i|\tau) = 1/k \sum_{s \in S_\tau} [P_\pi(u^i|s)]. \tag{26}$$

## B    Behavior Cloning and DAgger

Behavior Cloning is the simplest form of imitation learning, where learners imitate the demonstration of experts through supervised learning. Supervised learning requires the state-action pairs between experts and learners to be distributed i.i.d. However, in MDP, actions can affect the distribution of states. The incorrect actions of a learner lead to a deviation in state distribution, which is contrary to the i.i.d hypothesis.

Direct policy learning is an improved version of behavior cloning, represented by Data Aggregation (DAgger). To overcome the shortage of Behavior Cloning, DAgger requires the expert to provide feedback in the learners' trajectory rather than their own trajectory (interactive demonstration). Through interactive demonstration, the i.i.d hypothesis between learners and experts is guaranteed. However, the defect of DAgger is that experts need to perform interactive demonstrations in real time, which is impossible in some application scenarios.

## C    Pseudo-code

In this section, we describe the details of our algorithms, as shown in Algorithm 1.

---

**Algorithm 1** IGM structure with DAgger

---

   **Input:** Q-networks with global information $[Q^i(s, u^i)]_{i=1}^n$; decentralized Q-networks with local information $[q^i(\tau^i, u^i)]_{i=1}^n$; an experience replay buffer that stores past environment samples..
 1: **for** each learning episode of VD method **do**
 2:     Obtain initial state $s_0$ from environment
 3:     **for** t=0,...,until end of episode **do**
 4:         each learner agents take observation $[\tau_t^i]_{i=1}^n$
 5:         Select actions for each learner agents according to $[q^i(\tau^i, u^i)]_{i=1}^n$
 6:         Perform selected actions
 7:         Add environment sample $< s_t, s_{t+1}, [a_i^t]_{i=1}^N, r_t, [\tau_t^i]_{i=1}^n, [\tau_{t+1}^i]_{i=1}^n >$ into the replay buffer
 8:         **if** learning interval is reached **then**
 9:             Sample mini-batch $B$
10:             Train Q-networks with global information $[Q^i(s, u^i)]_{i=1}^n$ based on VD method using equation 6
11:             Train Q-networks with local information $[q^i(\tau^i, u^i)]_{i=1}^n$ based on supervised learning using proposition 5
12:         **end if**
13:     **end for**
14: **end for**

---

## D    Starcraft II environment setting and hyper-parameter

Table 2 shows the details of the Starcraft II environment setting, where the enemy units are controlled by the built-in AI. The task is to control ally units to defeat enemy units. The original observation range of ally units is nine. To fully demonstrate the robustness of our algorithm under lossy information, we reset the observation range to zero. A 24-core processor was used as a CPU, together with an Nvidia RTX3090 GPU.

In order to get a proper comparison effect, the parameters of all the algorithms to be tested are set to be the same. For optimization, all the algorithms use RMSProp with an alpha 0.99 and an epsilon 0.00001. The batch size used is 32. The experiment buffer size is 5000. The learning rate is 0.0005 for both the reinforcement learning part and the imitation learning part. In order to achieve good exploration, all algorithms use epsilon greedy action selection, with at least 0.1 exploration probability. Besides, all the algorithms are based on the double Q-learning algorithm to update the Q function, with a 200-update interval of target Q. The code of QMIX[†], QPLEX[‡] and DMIX[§] is referenced.

Table 2: SMAC challenges

| Map Name | Ally Units | Enemy Units |
| --- | --- | --- |
| 3s5z | 3 Stalkers & 5 Zealots | 3 Stalkers & 5 Zealots |
| 5m_vs_6m | 5 Marines | 6 Marines |
| MMM2 | 1 Medivac, 2 Marauders & 7 Marines | 1 Medivac, 3 Marauders & 8 Marines |
| 8m | 8 Marines | 8 Marines |
| 3s_vs_5z | 3 Stalkers | 5 Zealots |
| 8m_vs_9m | 8 Marines | 9 Marines |

## E    Additional experiment in single-agent environments

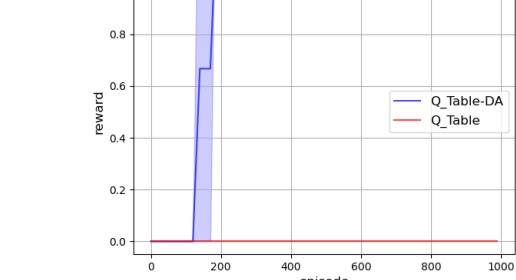

|  |  |
| --- | --- |
| SF FF | (S: start, safe) |
| FH FH | (F: frozen surface, safe) |
| FF FH | (H: hole, dead) |
| HF FG | (G: goal, target place) |

(a) Increased difficulty in frozenlake where each green box represents the same observation.

(b) DAgger based Q tabel get a feasible way.

Figure 6: Additional frozenlake experiment.

We further test the integrated DAgger structure in a simple single-agent environment - frozenlake. The frozenlake is a maze environment, and our task is to find a path from the starting position $S$ to the target place $G$ without staying in hole position $H$. If the agent reaches the target, it will get a reward of 1. As illustrated in Figure 6(a), the only safe positions are $S$ and $F$. To demonstrate the performance of the integrated DAgger structure, we make the experiment more difficult, so that the agent only knows which green box it was in but not its exact position. Compared with

---

[†] https://github.com/oxwhirl/pymarl
[‡] https://github.com/wjh720/QPLEX
[§] https://github.com/j3soon/dfac

the classical Q Table algorithm, we found that the Q Table algorithm could not learn a feasible strategy, while the integrated DAgger based Q Table method could get a path to the target. This is a preliminary experiment to show that integrated DAgger is expected to be extended to partially observed single-agent environments. Code is available in [¶] which is less than 100 lines.

# F   SMAC of nine sight view

The integrated DAgger structure is mainly used to solve the error accumulation of IGM-based hypernetwork methods. In the main text, we extensively test experiments of SMAC with 0 sight view. In this section, we will further test with a sight view of 9. As shown in Figure 7, we can find that the integrated DAgger method does not guarantee performance improvement. This is because, in the 9 sight view, global information is basically knowable. Without error accumulation, the integrated DAgger structure becomes redundant. In the 3s_vs_5z environment, the DAgger-based method degrades performance due to the extra training cost in imitation learning. In the 5m_vs_6m environment, the DAgger-based method outperforms the original method, and encouragingly, even a 0 observation range integrated DAgger-based method can produce similar performance with a 9 observation range original method.

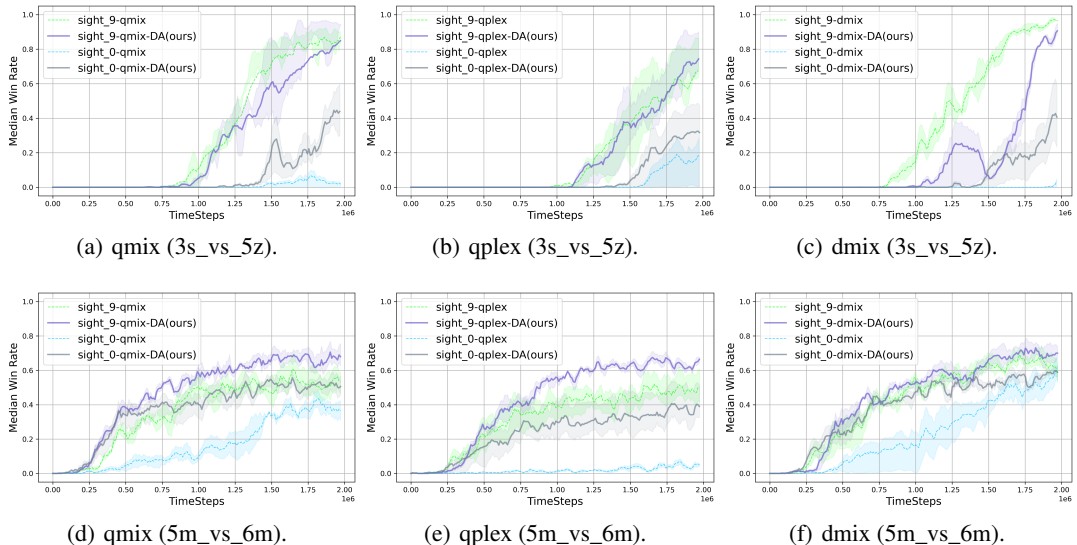

(a) qmix (3s_vs_5z).    (b) qplex (3s_vs_5z).    (c) dmix (3s_vs_5z).

(d) qmix (5m_vs_6m).    (e) qplex (5m_vs_6m).    (f) dmix (5m_vs_6m).

Figure 7: The limitation of integrated DAgger structure in nine observation range.

---