# OpenReview forum: "Rethinking Individual Global Max in Cooperative Multi-Agent Reinforcement Learning"
_NeurIPS.cc/2022/Conference — NeurIPS 2022 Accept_

### Official Review · Reviewer_yNmi · 2022-07-03

**Rating:** 6
**Confidence:** 4
**Soundness:** 3 good
**Presentation:** 3 good
**Contribution:** 3 good

**Summary:**

This paper rethinks a popular principle in cooperative MARL called Individual Global Max (IGM) decomposition, discovers the limitation of the IGM condition, and proposes an imitation learning strategy (IGM-DA) to address this problem. Due to partial observation, local observation may not provide enough information. Thus, the IGM condition will be a lossy decomposition. This paper trains an expert using a global state during centralized training and distils a decentralized agent using imitation learning. Empirical results show that IGM-DA can outperform state-of-the-art IGM-based approaches with zero sight view.

**Questions:**

For example, there are two landmarks to the left and right of the agents. Agents should go to the winning landmark. The probability of whether the left and right landmarks are winning landmark is equal. The local observations of each agent cannot see the location of the winning landmark. In this case, local history does not have enough information to make the right decisions. IGM condition is a lossy decomposition due to partial observation and can IGM-DA address this problem?

**Limitations:**

The main concerns are listed in the WEAKNESSES. The authors should discuss the examples presented in the reviewer's Question to justify IGM-DA.

**Strengths And Weaknesses:**

The reviewer will list the main strengths and weaknesses as follows:

STRENGTHS

IGM is a very popular principle in the value-based MARL algorithm. This paper rethinks the limitation of the IGM principle, which is very interesting and novel. Due to the partial observation, IGM is hard to keep the consistency between local and global greedy action selection. This paper proposes a teacher-student model to utilize the global information during centralized training to address this problem, which is an appropriate way to handle it.

WEAKNESSES

1) To address partial observation of IGM conditions, there is another way to enrich the observation of each agent using inter-agent communication [1]. By discussing this branch of related work, the position of this paper would become clearer.

2) Proposition 5 indicates that when multiple samples satisfy the same local observation $\tau$, imitation learning will take the average of these samples. The problem is that if the policies on these samples are completely different, imitation learning may lead to a bad policy, i.e., the policy distillation will fail due to insufficient information. The authors should discuss this case to justify the soundness of IGM-DA.

[1] Wang T, Wang J, Zheng C, et al. Learning nearly decomposable value functions via communication minimization[J]. ICLR 2020.

---

> ### Author Response · Authors · 2022-08-01
> **Thank you for your positive assessment!**
>
> > 1. To address partial observation of IGM conditions, there is another way to enrich the observation of each agent using inter-agent communication. By discussing this branch of related work, the position of this paper would become clearer.
>
> Thanks for drawing our attention to these relevant and interesting papers. Using extra communications is indeed an effective method to alleviate training instability caused by partial observations, provided that communication between agents is allowed. We briefly discussed these methods [1-3] in the newly added Section 5. Note, however, that the present work focuses on how to make agents learn a strategy more effectively in the presence of local observations without any communications between agents. In this regard, we found that IGM decomposition suffers from errors, and the training process will lead to error accumulation. To solve this problem, we integrated Dagger, an imitation learning technology, with IGM to construct a teacher-student model to reduce the overall error.
>
> [1] Wang T, Wang J, Zheng C, et al. Learning nearly decomposable value functions via communication minimization[J]. ICLR 2020.
> \
> [2] Foerster J, Assael I A, De Freitas N, et al. Learning to communicate with deep multi-agent reinforcement learning[J]. Advances in neural information processing systems, 2016, 29.
> \
> [3] Nguyen T T, Nguyen N D, Nahavandi S. Deep reinforcement learning for multiagent systems: A review of challenges, solutions, and applications[J]. IEEE transactions on cybernetics, 2020, 50(9): 3826-3839.

---

> > ### Author Response · Authors · 2022-08-01
> > **Thank you for your positive assessment!**
> >
> > > 2. Proposition 5 indicates that when multiple samples satisfy the same local observation $\tau$, imitation learning will take the average of these samples. The problem is that if the policies on these samples are completely different, imitation learning may lead to a bad policy, i.e., the policy distillation will fail due to insufficient information. The authors should discuss this case to justify the soundness of IGM-DA.
> >
> > > For example, there are two landmarks to the left and right of the agents. Agents should go to the winning landmark. The probability of whether the left and right landmarks are winning landmark is equal. The local observations of each agent cannot see the location of the winning landmark. In this case, local history does not have enough information to make the right decisions. IGM condition is a lossy decomposition due to partial observation and can IGM-DA address this problem?
> >
> > In our teacher-student model, only the learners interact with the environment to generate trajectories. The experts evaluate the learners' trajectories and propose improvements, which are then absorbed by the learners in the form of Proposition 5. The new strategy of the learner is always based on the local information only. This new strategy then continues to interact with the environment. The whole training process is equivalent to a closed-loop control, which has two important components: global information-based reinforcement learning training and Dagger, a real-time strategy distillation. For the actual environment and a current strategy, we need to consider two questions: is the current strategy a locally optimal solution? Can this optimal strategy be distilled well? The current strategy will continue to improve or explore through ε greedy until convergence.
> > - For problem:
> > The policy distillation will fail due to insufficient information. The authors should discuss this case to justify the soundness of IGM-DA.
> >
> > Answer:
> > If the policy distillation fails due to insufficient information, learners will return the failed strategy to experts for further revision and experts will continue to adjust the strategy or explore through ε greedy until a suitable strategy is found.
> >
> > Take the scenario you mentioned as an example. For a single agent, because one does not know which side is the winning landmark, any left-to-right strategy is reasonable. In case two agents are cooperating, the optimal strategy is to explore both left and right. Because our method uses global information for reinforcement learning, this optimal strategy will be discovered without requiring any observations. Thus, tis optimal strategy can also be learned by the learners.
> > Consider the traditional method, on the contrary, the agents try to iteratively learn an action value based on local observations through the Bellman equation. However, the local observation of the whole environment remains unchanged, leading to fluctuations in the iterative process. That is, the calculation of action value based on local information will become instable. This is also the reason why we use global information for value decomposition - to avoid unstable training (error accumulation) caused by local observations. Please refer to Appendix E, in which we present our simple experiments under a single agent, which also prove the advantages of our method against partial observations.
> >
> > - For problem:
> > IGM condition is a lossy decomposition due to partial observation and can IGM-DA address this problem?
> >
> > Answer:
> > IGM condition is a lossy decomposition due to partial observation, which will lead to error accumulation during the bellman iterative training. IGM-DA avoids introducing partial observation into IGM decomposition, which is lossless and does not produce cumulative error in the training process.

---

> > > ### Author Response · Authors · 2022-08-08
> > > **Awaiting your reply!**
> > >
> > > I look forward to your other questions and suggestions.

---

### Official Review · Reviewer_bd75 · 2022-07-10

**Rating:** 6
**Confidence:** 4
**Soundness:** 1 poor
**Presentation:** 1 poor
**Contribution:** 3 good

**Summary:**

This paper analyzes the error accumulation in multi-agent Q-learning and focuses on the error caused by partial observability. Regarding this problem, this paper integrates a DAgger module to distill the full-observability policy to the partial-observability policy.

**Questions:**

Major concerns:

1. What is the **formal definition** of $error$-terms in proposition 2? Does it mean Bellman errors? What is $Error(Q)$? These error terms are the core concepts of this paper and should be defined in formal mathematical formulas.

2. There are no experiment results supporting the error accumulation is reduced from Eq.(7) to Eq.(9) in practice. It raises a concern that the performance improvement may come from any unmentioned implementation details.

3. The loss function of "supervised imitation learning" should be included in the main text. I even cannot find it in the appendix.

4. What is the purpose of proposition 5. I do not understand what "satisfy local observation" means in proposition 5 and what is "the optimal action-value after imitation learning".

Minor:

1. Discount factor is missing in Eq.(7).

2. In Eq.(10), please use the indicator function to represent the one-hot probability.

**Limitations:**

The discussions on background literature and related work are limited. I do not penalize my evaluation by this point, but I strongly encourage authors to include a related work section in the paper.

As this paper is titled by "rethinking IGM", it would be better to connect with other papers studying the algorithmic property of VD methods (maybe in other aspects), e.g., [1-3].

[1] Castellini et al. "The Representational Capacity of Action-Value Networks for Multi-Agent Reinforcement Learning" AAMAS 2019.

[2] Wang et al. "Towards Understanding Cooperative Multi-Agent Q-Learning with Value Factorization" NeurIPS 2021.

[3] Huang et al. "Multiagent Q-learning with Sub-Team Coordination" AAMAS 2022.

**Strengths And Weaknesses:**

Strength: This paper studies an important problem and the idea is novel and makes sense.

Weakness: Although I can roughly capture the main idea of this paper, the notation and definition of many core concepts are not clear (see questions). The writing quality is far from a conference publication.

---

> ### Author Response · Authors · 2022-08-01
> **Thank you very much for your valuable comments, which have helped us to improve the quality of this manuscript.**
>
> > 1. What is the formal definition of error-terms in proposition 2? Does it mean Bellman errors? What is $Error(Q)$? These error terms are the core concepts of this paper and should be defined in formal mathematical formulas.
>
> This error term represents the current error in action values. Assuming that the real action value of the current strategy is  $Q^\pi$, and the calculated action value is  $\hat{Q}^\pi$, then the error term is defined as $Error(Q^\pi )=Q^\pi -\hat{Q}^\pi$. We have added a definition of the error term before proposition 2 in the paper. Error accumulation discussed in the paper comes from the iterations of the bellman equation based on partial information. In this work, we avoid carrying out bellman equation iterations on the basis of partial information, thus avoiding error accumulation.
>
> > 2. There are no experiment results supporting the error accumulation is reduced from Eq.(7) to Eq.(9) in practice. It raises a concern that the performance improvement may come from any unmentioned implementation details.
>
> Unfortunately, it is difficult to experimentally show the errors in action values, which requires to know the real action value of a given strategy, and the calculation of the real correct action value relies on a complete understanding of the environment. Nevertheless, extensive empirical results demonstrated the robust performance of the proposed algorithm for scenarios with partial observations. The details of the proposed algorithm can be found in Appendix D. In addition, we also provide a simple implementation of the algorithm in a single agent environment with partial observation. The code has less than 100 lines, which is convenient for the reader to verify.
>
> > 3. The loss function of "supervised imitation learning" should be included in the main text. I even cannot find it in the appendix.
>
> Apologies for the missing definition of the loss function. We have included the loss function of the supervised imitation learning (as shown below) in Proposition 5 of the revised manuscript.
> $loss = {q^i}({\tau ^i},{\mu ^i}) - 1/k\sum\nolimits_s {[{P_\pi }({\mu ^i}|s)]} $
>
> > 4. What is the purpose of proposition 5. I do not understand what "satisfy local observation" means in proposition 5 and what is "the optimal action-value after imitation learning".
>
> This is an important part of our method. Assume we have obtained the individual action value provided by a group of experts. Our next task is to extract the action value of the learners from the action value of experts. According to the minimum risk Bayes decision rule, for a selected action, its risk can be defined as difference between the optimal action value of the expert and the currently selected action value. However, the individual action value has no practical significance because of the existence of IGM. The suboptimal individual action value of the experts does not necessarily mean the suboptimal individual action (IGM only guarantees that the maximum local action value corresponds to the maximum joint-action value. However, the secondary local action value does not necessarily correspond to the secondary joint-action value). Thus, a more practical definition of the risk is a penalty of 1 if the current action selection is inconsistent with the optimal action of the expert label, and no penalty if consistent. In other words, the minimum risk Bayes decision rule degenerates into the minimum error rate Bayes decision rule. Therefore, when we calculate the estimated risk of each action, the optimal action selection is the action with the minimum risk.
>  -  "satisfy local observation"
>
> Because we need to estimate the risk term by sampling, “satisfy local observation $\tau$” is the condition of selecting samples, see the proof of Proposition 5 in Appendix A.
> - "the optimal action-value after imitation learning"
>
> According to the minimum risk Bayes decision rule, we first define the expected loss and penalty functions. Because the action with the minimum expected risk (error) is the most popular one, we can define the action value after imitation learning as the negative expected risk. We recognize that 'best' here may be misleading, therefore, we have deleted it in the revised paper.

---

> > ### Author Response · Authors · 2022-08-01
> > **Thank you very much for your valuable comments, which have helped us to improve the quality of this manuscript.**
> >
> > > Minor: \
> > &#8194; Discount factor is missing in Eq.(7).  \
> > &#8194; In Eq.(10), please use the indicator function to represent the one-hot probability.
> >
> > Thank you very much for spotting the typos, which have been corrected.
> >
> > > Limitations: \
> > The discussions on background literature and related work are limited. I do not penalize my evaluation by this point, but I strongly encourage authors to include a related work section in the paper.
> >
> > We thank you for your constructive comments. According to your suggestion, we have added a new section (Section 5, “Related work”) discussing the most relevant work, including the following papers [1-3].
> >
> > [1] Castellini et al. "The Representational Capacity of Action-Value Networks for Multi-Agent Reinforcement Learning" AAMAS 2019. \
> > [2] Wang et al. "Towards Understanding Cooperative Multi-Agent Q-Learning with Value Factorization" NeurIPS 2021. \
> > [3] Huang et al. "Multiagent Q-learning with Sub-Team Coordination" AAMAS 2022.

---

> > > ### Author Response · Authors · 2022-08-08
> > > **Awaiting your reply!**
> > >
> > > I look forward to your other questions and suggestions.

---

### Official Review · Reviewer_So3D · 2022-07-11

**Rating:** 5
**Confidence:** 4
**Soundness:** 1 poor
**Presentation:** 3 good
**Contribution:** 1 poor

**Summary:**

This work analyzes the current IGM decomposition framework in CTDE problem, especially the lossy in value decomposition and the accumulation of lossy in the training process. Based on the analysis, this work proposes a method combined with imitation learning to solve the lossy between global observation and local observation of Q (s, U) = \sum Q(s_i, u_i) in value decomposition.
The method part includes an expert agent trained using off-policy algorithm with global observation and a learner agent trained using supervised algorithm with local observation,  through alternately training, the training lossy of learner is alleviated.


**Questions:**

In CTDE, the decentralized execution is constrained with partial observation, the difference between global obs and local obs is determined by the partial obs environment, this problem can not be solved by imitating an expert. I'm not sure it's always good to imitate global state expert, especially, when the learner agent needs to execute independently.

I think some previous works have proposed some methods like using a CTCE teacher and a CTDE student, what's the difference when comparing this paper to these works?

**Ethics Review Area:**

["I don’t know"]

**Limitations:**

This paper analyses the value decomposition error problem, but the method of this paper is of limited help in solving this problem.

**Strengths And Weaknesses:**

Strength:
The paper is well organized and well written and the analysis/method part is very detailed, and the definition is more formal.
It is a relatively novel perspective to integrate imitation learning into the framework of CTDE  to alleviate value decomposition error.

Weakness:
The flaws of IQM-based method is widely studied. This paper has spent a lot of space on analysis, but most of the analysis is not deep enough and I think the analysis result is not innovative enough.
The experiment part is insufficient. The setting of the experiment is zero sight SMAC, which is obviously more conducive to the setting of this method, I think adding some experiments with normal sight in SMAC will be better.

---

> ### Author Response · Authors · 2022-08-01
> **Thank you for your valuable review !**
>
> > In CTDE, the decentralized execution is constrained with partial observation, the difference between global obs and local obs is determined by the partial obs environment, this problem can not be solved by imitating an expert. I'm not sure it's always good to imitate global state expert, especially, when the learner agent needs to execute independently.
>
> As the reviewer correctly pointed out, CTDE is by far the most popular multi-agent reinforcement learning paradigm, and many empirical results have demonstrated its effectiveness. Although we agree that the issue of missing global observation cannot be completely resolved by imitating an expert, we show that learning from an expert is able to improve the performance. In particular, this work focuses on addressing the error accumulation problem, which has not been considered in the literature. In the proposed algorithm, Dagger is an important part that requires experts to evaluate and improve learner strategies rather than their own strategies. As a result, we can get a strategy depending on the local information, thereby avoiding error accumulation.
>
> > I think some previous works have proposed some methods like using a CTCE teacher and a CTDE student, what's the difference when comparing this paper to these works?
>
> To the best of our knowledge, it is the first time that both IGM decomposition error and error accumulation have been considered. By seamlessly coupling IGM with the Dagger algorithm, we avoid error accumulation in IGM decomposition resulting from local observations without requiring human experts.
>
> Regarding similar previous work, we found one article using CTCE and distillation learning [1]. The research, however, assumes that communications between agents are allowed. The main focus of that paper introduced completely centralized training to address the low efficiency of sampling resulting from frequent changing communication topology. Compared with our research, the motivations are completely different. We provided a brief discussion of this work in a newly added Section 5 (“Related work”).
>
> > Limitations:
> This paper analyses the value decomposition error problem, but the method of this paper is of limited help in solving this problem.
>
> We have achieved an average improvement of 20 % over the state-of-the-art algorithms in the SC2 experiment with a sight view of 0. And we have conducted new experiments when the sight view is 5 and 9, respectively, to prove the robustness of our method. We also extended our structure to partially observed single-agent environments (Appendix E), which also showed promising performance. The code of the single agent experiment has less than 100 lines, which is also fairly convenient for the reader to verify.
>
>
> [1] Chen G. A New Framework for Multi-Agent Reinforcement Learning–Centralized Training and Exploration with Decentralized Execution via Policy Distillation[C]//Proceedings of the 19th International Conference on Autonomous Agents and MultiAgent Systems. 2020: 1801-1803.

---

> > ### Author Response · Authors · 2022-08-08
> > **Awaiting your reply!**
> >
> > I look forward to your other questions and suggestions.

---

> > ### Comment · Reviewer_So3D · 2022-08-08
> > **the IGM decomposition error and error accumulation**
> >
> > I think error accumulation is resulting from IGM decomposition error, so when we replace $q(\tau, u)$ with $q(s, u)$, the error term will disappear, why the $error_{dec}$ of time step t is left in Eq.(9)?

---

> > > ### Author Response · Authors · 2022-08-08
> > > **Thank you for your reply!**
> > >
> > > I am always glad to receive your reply. Eq.(9) analyzes the overall error in our framework. By replacing $q(\tau,u)$ with $q(s,u)$, we can avoid the partial observation error $error_{dec}$ being accumulated in the Bellman iteration training process. However, in the end, our strategy needs to be applied to partially observed scenarios, which will still produce errors due to partial observation (of course, the error accumulation is avoided). You can refer to Figure 2 (a), in the second stage, we train the individual learner agent based on local observation by means of imitation learning.
> > >
> > > This work focuses on addressing the error accumulation problem and proposes to introduce imitation learning into VD method.

---

> > > ### Author Response · Authors · 2022-08-09
> > > **Thank you for your reply. I hope this clears up your doubts.**
> > >
> > > Because the $error_{dec}$ caused by partial observation is inevitable, this is the reason why $error_{dec}$ of time step t is left in Eq.(9). However, by comparing equation 7 and equation 9, we can see that error accumulation resulting from $error_{dec}$ can be avoided.

---

### Official Review · Reviewer_7jPS · 2022-07-12

**Rating:** 6
**Confidence:** 4
**Soundness:** 3 good
**Presentation:** 2 fair
**Contribution:** 3 good

**Summary:**

This paper provides a new perspective on the study of Q-value decomposition methods in the field of multi-agent reinforcement learning. In order to maintain decentralized execution while avoiding learning non-stationarity during the training phase, centralized training with decentralized execution paradigm is widely studied. The authors pinpoint a gap between local and global observation, which leads to the fact that local action-value functions cannot represent the right Q values. Such a representational error may accumulate temporally. The authors use a supervised learning method (with the "label" being action values conditioned on global observations) to solve this problem and found it can improve the performance of multiple MARL methods.

**Questions:**

(1) It is not surprising that local Q functions conditioned on local information are not accurate enough. Can the authors give a bound to $Error_{dec}$? Perhaps by quantifying the ratio of lossy information in $\tau$. (It is fine if this is too difficult, but the reviewer will fight for the acceptance of this paper if the author can make it and change Figure 3.)

(2) Figure 3 does not give many insights. It should be replaced. Please demonstrate the gap between Q functions depending on local and global information.

(3) Please show the influence of the sight range (changing from 0 to, say, 5). For now, a zero sight range may be too extreme.

**Limitations:**

The reviewer thinks the paper has made many interesting findings and does not see obvious negative societal impacts

**Strengths And Weaknesses:**

### Significance

The paper discusses an interesting issue in CTDE value-based multi-agent learning methods that are ignored in the previous literature. As far as the reviewer is concerned, the paper is interesting, and the contribution may inspire many MARL researchers.

### Quality
(1) The major concern of the reviewer is that the experimental results cannot fully support the analysis in the previous sections. Figure 3 does not make sense. A smaller sight range must hurt the performance, but it may simply be because they have little information about the other agents and enemies. It cannot be guaranteed that the discussed issue in this paper leads to this performance gap.

(2) Another drawback of the paper is that some results are not surprising. For example, it is well-known that Q functions conditioned on local observations are less accurate than those based on global information. And such an error will certainly accumulate temporally. It would be much more interesting if the authors could bound this error.

---

> ### Author Response · Authors · 2022-08-01
> **Thank you for your positive assessment!**
>
> > (1) It is not surprising that local Q functions conditioned on local information are not accurate enough. Can the authors give a bound to $Error_{dec}$? Perhaps by quantifying the ratio of lossy information in $\tau$. (It is fine if this is too difficult, but the reviewer will fight for the acceptance of this paper if the author can make it and change Figure 3.)
>
> Thanks for your appreciation of our work and your insightful comments. We fully agree that providing a bound of the error is of particular interest. Although your suggestion to quantifying the ratio of lossy information is an attractive idea, we are not able to provide a theoretic bound of the error. It is interesting to note that, in reinforcement learning, the calculation of action value (strategy generation) depends on the reward function and Markov property of the environment. The dependence of the reward function and Markov property on global information will seriously affect the calculation of action value (strategy generation). However, the obtained strategy does not necessarily have such strong dependence on global information. Taking the maze as an example, a simple strategy is to walk right, which does not depend on any position information. In short, the quantitative analysis of action value error is a complex problem. If the quantitative analysis of action value error can be realized, it is expected to provide reference for the number of sensors in practical engineering.
>
> > (2) Figure 3 does not give many insights. It should be replaced. Please demonstrate the gap between Q functions depending on local and global information.
>
> In proposition 1, we only prove that if the observation is incomplete, the performance of some complex strategies (Q function) will be affected. For a real environment, we still need to carry out experiments to empirically verify this, which is the purpose of the experimental results presented in Figure 3 in the previous manuscript. Of course, as the reviewer pointed out, comparing the winning rate only may be insufficient to demonstrate the gap between the Q functions when global information is available or not. Consequently, we conducted new experiments to illustrate the differences.
>
> In the presence of partial observations, different global states may correspond to the same local observations, and the action value of the learner under a local observation is the weighted average of the action value of the expert under different global observations. Figure 3 (b) in the revised manuscript shows the distribution of the action values. The discrepancies in the action distribution may eventually lead to wrong action choices of the learners.
>
> > Please show the influence of the sight range (changing from 0 to, say, 5). For now, a zero sight range may be too extreme.
>
> Following your suggestion, we added an experiment with an additional sight view of 5 for the Qmix algorithm and the scene of 5 m _ vs _ 6 m. As shown in Fig. 5 (c), our algorithm is more robust to partial observations.

---

> > ### Comment · Reviewer_7jPS · 2022-08-05
> > **Thanks for your response.**
> >
> > New Figure 3 shows Q functions at two time steps. The reviewer thinks it is not convincing. Why not plot curves showing Q differences (averaged over all timesteps and actions) over time.
> >
> > New Figure 5(c) is quite interesting. DAgger with sight range 9 >  DAgger with sight range 5 >  DAgger with sight range 0. However, the proposed method is expected to address the problem of lossy decomposition, and these gaps between DAgger with different sight ranges need further elaborations.

---

> > > ### Author Response · Authors · 2022-08-06
> > > **Thank you for your positive assessment!**
> > >
> > > According to your suggestion, we have plot curves showing Q differences (averaged over all timesteps and actions) over time. You can browse anonymous links to view the image (https://anonymous.4open.science/r/pymarl_HDA-5726/picture/Q%20Difference.png).
> > >
> > > More experiments on sight view 9 are available in Appendix F. Moreover, we also extend simple experiment in the single agent environment in Appendix E, which also proves the advantages of our method.

---

> > > > ### Comment · Reviewer_7jPS · 2022-08-07
> > > > **Thanks for your response.**
> > > >
> > > > Thank the authors for their responses. I tend to keep my score.

---

### Meta-Review · Area_Chair_z1ye · 2022-08-22

**Recommendation:** Accept
**Confidence:** Less certain

**Metareview:**

This paper revisits the notion of Individual Global Max in multi-agent reinforcement learning, in particular considering how to address the fact that individual greedy actions may not be globally optimal in cooperative settings.

Overall, the general sentiment is that this is interesting work with a useful contribution, but that the paper could be further improved. There were some specific concerns regarding the experimental results, which the authors answered in the rebuttal. The results for sight view 5 are particularly relevant, given that they identify a setting in which the system is not extremely partially observable, but where their algorithm still provides benefits.

I also note that the paper's presentation is less polished than it could be in a number of places (For example, Table captions are sometimes brief / missing punctuation, graphs are somewhat hard to read, the equation in Prop. 5 should be indented).

**Award:**

No

---

### Decision · Program_Chairs · 2022-09-14

Accept